# Forcing Mechanisms of the Interannual Sea Level Variability in the Midlatitude South Pacific during 2004–2020

**C. Germineaud [1,2,3,4,]**[*], **D. L. Volkov [1,2]**, **S. Cravatte [3] and W. Llovel [5]**

[1] Cooperative Institute for Marine and Atmospheric Studies, University of Miami, Miami, FL 33149, USA
[2] NOAA, Atlantic Oceanographic and Meteorological Laboratory, Physical Oceanography Division, Miami, FL 33149, USA
[3] LEGOS, Université de Toulouse, CNES, CNRS, IRD, UPS, 31400 Toulouse, France
[4] Centre National d'Etudes Spatiales, 31401 Toulouse, France
[5] LOPS, CNRS, University of Brest, IFREMER, IRD, 29238 Plouzané, France
[*] Correspondence: cyril.germineaud@cnes.fr

**Abstract:** Over the past few decades, the global mean sea level rise and superimposed regional fluctuations of sea level have exerted considerable stress on coastal communities, especially in low-elevation regions such as the Pacific Islands in the western South Pacific Ocean. This made it necessary to have the most comprehensive understanding of the forcing mechanisms that are responsible for the increasing rates of extreme sea level events. In this study, we explore the causes of the observed sea level variability in the midlatitude South Pacific on interannual time scales using observations and atmospheric reanalyses combined with a 1.5 layer reduced-gravity model. We focus on the 2004–2020 period, during which the Argo's global array allowed us to assess year-to-year changes in steric sea level caused by thermohaline changes in different depth ranges (from the surface down to 2000 m). We find that during the 2015–2016 El Niño and the following 2017–2018 La Niña, large variations in thermosteric sea level occurred due to temperature changes within the 100–500 dbar layer in the midlatitude southwest Pacific. In the western boundary region (from 30°S to 40°S), the variations in halosteric sea level between 100 and 500 dbar were significant and could have partially balanced the corresponding changes in thermosteric sea level. We show that around 35°S, baroclinic Rossby waves forced by the open-ocean wind-stress forcing account for 40 to 75% of the interannual sea level variance between 100°W and 180°, while the influence of remote sea level signals generated near the Chilean coast is limited to the region east of 100°W. The contribution of surface heat fluxes on interannual time scales is also considered and shown to be negligible.

**Keywords:** South Pacific Ocean; interannual sea level variability; surface forcing mechanisms; upper-ocean warming

## 1. Introduction

Global mean sea level (GMSL), derived from satellite altimetry, has been rising at a rate of 3.4 mm yr$^{-1}$ (e.g., [1,2]) over the past three decades due to increasing atmospheric greenhouse gas concentrations and the related Earth's energy imbalance (e.g., [3,4]). Most of the excess thermal energy (>90%) in the climate system has been stored in the world's oceans (e.g., [5,6]), resulting in the GMSL rise due to ocean warming via thermal expansion [7]. In addition to ocean warming, the GMSL rise is caused by inputs of freshwater due to the melting of icesheets and mountain glaciers (e.g., [8]). Superimposed on the GMSL rise, several regions experience changes in sea level with rates greater than the GMSL rise, as a response to changes in ocean circulation and atmospheric forcing over different time scales, exerting extensive stress on low-lying coastal zones (e.g., [7]). This is notably the case in the South Pacific Ocean, where several coastal communities are facing accelerated sea level rise at rates that are 3 to 4 times higher than the GMSL rise (e.g., [9,10]). Moreover, this region (hosting small islands) has been identified as vulnerable to coastal flooding

in response to future sea level rise (e.g., [11]). Previous studies have emphasized that interannual sea level variations in the South Pacific are essentially due to steric fluctuations (e.g., [12,13]), resulting from temperature and salinity changes in the ocean. Changes in the thermosteric (temperature-related) component only are the main contributor of sea level change in the South Pacific at these time scales (e.g., [14,15]). Nevertheless, contributions from the halosteric (salinity-related) component have been shown to play a sizable role in several areas, such as the western equatorial Pacific and near the western boundary currents (e.g., [13,15,16]). In the southeast Indian Ocean, large halosteric contributions to interannual and multidecadal variabilities of sea level were also found [17,18].

The drivers of these interannual steric variations have also been investigated. In the tropical Pacific, the interannual variability in steric sea level (SSL), and both its thermosteric and halosteric sea level components (TSL and HSL, respectively) have been shown to be essentially driven by El Niño Southern Oscillation (ENSO) (e.g., [15,19,20]), via changes in wind-stress forcing and surface buoyancy fluxes. At the mid-to-high latitude South Pacific, the impact of ENSO on sea level change via mixed teleconnections (see [20]) is less straightforward, partly due to the influence of another leading mode of regional climate variability, the Antarctic Oscillation (AAO), also known as the Southern Annular Mode (see [21]). Past studies (e.g., [22,23]) have shown that the interannual ocean circulation variability at midlatitudes in the South Pacific is influenced by the AAO forcing (intensified/weakened westerly winds). This is also the case near the western boundary, south of 30°S in the Tasman Sea, west of New Zealand (Figure 1a), which is home of the eddy-rich East Australian current (EAC). Some studies have also shown that a sizable fraction of the interannual sea level variability and its steric component can be explained by intrinsic ocean variability south of about 20°S in the South Pacific [12,24]; particularly, in the vicinity of the EAC. This is consistent with Cravatte et al. [25], who identified that the interannual transport variability of the EAC was predominantly influenced by nonlinear (chaotic) ocean processes. Moreover, it is noteworthy that decadal and long-term sea level variations in the South Pacific near New Zealand were shown to be primarily driven by changes in wind-stress forcing associated with the Interdecadal Pacific Oscillation (IPO) and the AAO, respectively [26].

During the 2005–2014 decade, the upper-ocean (from the surface down to 2000 m) heat content (HC) was increasing in the subtropical southwest Pacific region (e.g., [27–29]), contributing to about a quarter of the global mean SSL rise [30]. This increase in ocean HC and related TSL has been attributed to convergence-favorable wind-stress forcing in the subtropics and resultant downward Ekman pumping [29,30]. Indeed, intensified southeasterly trade winds (typically observed during La Niña-like conditions) in the tropics associated with strong westerly winds south of 35°S (characteristic for the positive phases of the AAO; [21]) in the South Pacific were responsible for the wind-driven Ekman convergence during 2005–2014. This led to downward heaving of isotherms and the accumulation of heat in the region of 25°S–50°S, 130°W–180° over the upper 2000 m, but possibly also at depths greater than 2000 m [30]. It is thus possible that the interannual sea level variability in the South Pacific region south of 25°S arises from the interplay between ENSO and AAO wind forcing changes. In addition to purely local effects of wind forcing, Qiu and Chen [31] showed that wind-forced sea level anomaly (SLA) signals propagating westward as Rossby waves along 40°S in the South Pacific largely explained the interannual variability of sea level from October 1992 through 2004. More recently, one of the strongest on record El Niño events occurred in 2015–2016 (e.g., [32]), followed by a rather weak La Niña in 2017–2018 (e.g., [33]). The two events induced large redistributions of heat and associated sea level changes in the tropical Pacific and the midlatitude South Pacific.

The main objective of this study is to better understand the key drivers (local and remote) of the interannual sea level variability in the midlatitude South Pacific for the entire period of 2004–2020. First, to complement the previous studies that identified the wind-driven Ekman convergence (and resultant downward isopycnal heaving) as the dominant driver of sea level change in the South Pacific during 2005–2014, we analyze the

basin-scale impact of local wind forcing (through Ekman pumping) on temperature and salinity contributions (from the surface to 2000 dbar) to interannual sea level variability over 2015–2020. We also analyze the role of baroclinic Rossby waves (i) generated near the eastern boundary (i.e., the Chilean coast), (ii) forced by wind-stress curl changes over the open-ocean, and (iii) forced by changes in surface heat fluxes. For this purpose, we employ a 1.5-layer reduced-gravity model (hereinafter RGM)—an approach that has been frequently used in the past to study the variability of sea level and ocean circulation in the South Pacific (e.g., [10,31,34,35]), and in the South Indian Ocean (e.g., [36–38])—complemented with a heat flux forcing term (e.g., [39]).

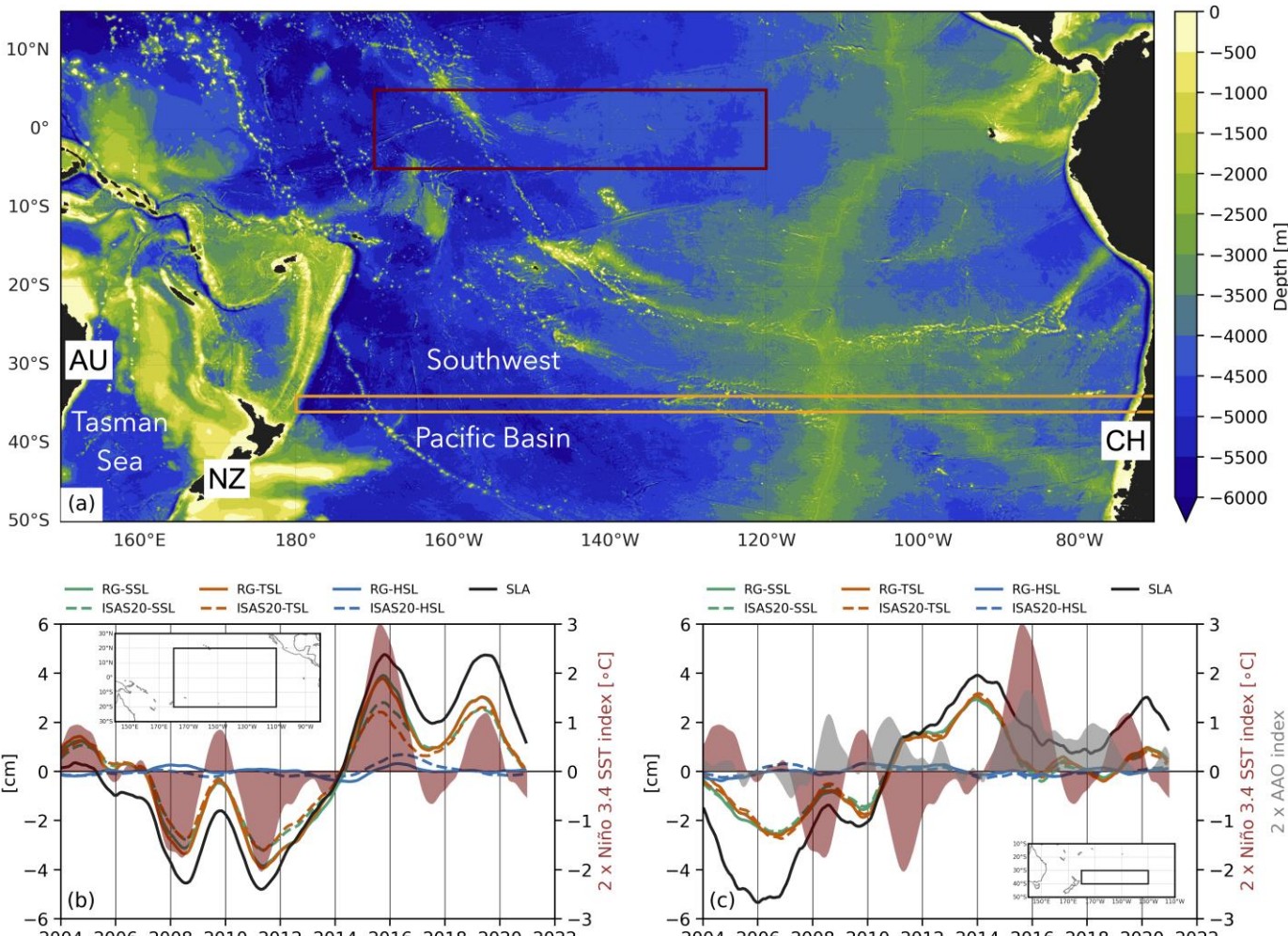

**Figure 1.** (**a**) Bathymetry map derived from the gridded 15 arc-second General Bathymetric Chart of the Oceans (GEBCO) 2022. Country names are indicated on the map as follows: Australia (AU); New Zealand (NZ) and Chile (CH). The dark red rectangle shows the Niño 3.4 SST index region, while the orange rectangle shows the region of the reduced-gravity model used in this study. (**b,c**) Interannual sea level budget with altimetry sea level anomaly (SLA; black) and 0–2000 dbar Argo steric contributions during 2004–2020 and averaged over (**a**) 20°S–20°N, 180°–110°W and (**b**) 30°S–40°S, 180°–130°W (black rectangles in the insets for geographic location). In (**a,b**), time series of SLA (black), steric sea level (SSL; green), thermosteric sea level (TSL; orange) and halosteric sea level (HSL; blue) are shown. Argo contributions derived from the Roemmich-Gilson (RG) Climatology and the 8th version of the In Situ Analysis System (ISAS20) are, respectively, in solid and dashed lines. Red and gray shadings represent the Niño 3.4 sea surface temperature (SST) index and the AAO index, respectively, with a 1.5 year running mean applied and scaled by 2 for plotting purposes.

## 2. Data and Models

This study covers the Argo era starting from 2004, when the Argo array reached a nearly global extent. To focus only on interannual time scales, all data used in this study was processed as follows: (i) the seasonal cycle was removed by subtracting the monthly mean climatology over 2004–2020 and (ii) the data was low-pass filtered using a running mean with a 1.5-year cutoff period.

### 2.1. Altimetric Sea Level Data

To investigate the mechanisms of dynamic sea level changes over the Argo period of 2004–2020, we used monthly $0.25° \times 0.25°$ gridded fields of SLA derived from satellite altimetry and distributed by CMEMS (the Copernicus Marine Environment Monitoring Service). The SLA fields used in this study correspond to the CMEMS delayed-time monthly gridded product identified as SEALEVEL_GLO_PHY_L4_MY_008_047 (https://doi.org/10.48670/moi-00148). This monthly product of SLA was calculated relative to a twenty-year mean (1993–2012) and generated using data from all satellite missions available at the time (Jason-3, Sentinel-3A, HY-2A, Saral/AltiKa, Cryosat-2, Jason-2, Jason-1, Topex/Poseidon, ENVISAT, GFO, ERS1/2), including instrumental and geophysical corrections (e.g., [40]). To focus on the dynamic sea level variability (driven by ocean and atmospheric dynamics) in the South Pacific Ocean, the GMSL over 2004–2020 was subtracted from the SLA time series at each grid point.

### 2.2. Atmospheric Reanalyses and Climate Indices

To relate the effects of local wind forcing (via Ekman pumping) to sea level changes, we use monthly $0.25° \times 0.25°$ gridded fields of surface (10-m) wind speed and wind-stress ($\boldsymbol{\tau}$) during 1993–2020 from the ECMWF's (European Centre for Medium-Range Weather Forecasts) ERA5 atmospheric reanalysis [41]. The Ekman pumping anomalies ($W_{Ek}$; positive upward) are calculated as follows:

$$W_{Ek} = \nabla \times \frac{\boldsymbol{\tau}'}{\rho_0 f} \tag{1}$$

where $\boldsymbol{\tau}'$ represents the wind-stress anomaly (relative to the period of 1993–2020), $\rho_0 = 1025 \text{ kg m}^{-3}$ and $f$ (in $\text{s}^{-1}$) is the Coriolis parameter. The sea level change due to local wind forcing through the time integral of $W_{Ek}$ ($\Delta h_{Ek}$) is calculated using the following Ekman solution:

$$\Delta h_{Ek} = \frac{g'}{\rho_0 g f} \int_{t1}^{t2} \nabla \times \boldsymbol{\tau}' dt \tag{2}$$

where $g'$ and $g$ are, respectively, the reduced gravity and the gravity (in $\text{m s}^{-2}$); $t_1$ and $t_2$ are the lower- and upper-time limits of integration. Note that $g'$ is based on values derived by Qiu and Chen ([31]; their Figure 5) from 50°S to 10°S in the South Pacific and then linearly extrapolated in the tropical regions of 5°–10°S and 5°–15°N.

The monthly $0.25° \times 0.25°$ gridded fields of surface net shortwave and longwave radiations from ERA5, and the corresponding fields of surface latent and sensible heat fluxes are summed to estimate the net surface heat fluxes, $Q_{net}$ from 2004 to 2020. For comparison purposes, $Q_{net}$ was also estimated from the National Centers for Environmental Prediction and the National Center for Atmospheric Research (NCAR) Reanalysis 1 (hereinafter NCEP1; [42]) and the Japanese Meteorological Agency 55-year Reanalysis (hereinafter JRA55; [43]) products. Note that $Q_{net}$ fields derived from NCEP1 ($2.5° \times 2.5°$ monthly grids) and JRA55 ($1.25° \times 1.25°$ monthly grids) for the 2004–2020 period were interpolated to the ERA5 $0.25° \times 0.25°$ grid prior to computing the relative contribution of net surface heat fluxes to interannual sea level variations (see Section 2.4). Air-sea fluxes from atmospheric reanalysis datasets are well-known to show a significant spread in the tropical and subtropical regions, notably because of differences in bulk flux parameterizations (e.g., [44]). This implies that discrepancies between the surface heat flux

contributions derived from ERA5, NCEP1 and JRA55 can be observed; however, the results presented in Section 3.3 will suggest otherwise.

To characterize ENSO events, we use the Niño 3.4 sea surface temperature (SST) index calculated by averaging the SST anomalies (with respect to the 1981–2010 time mean) in the Equatorial Pacific region of 5°S–5°N, 120°W–170°W (dark red box in Figure 1a), from the National Oceanic and Atmospheric Administration (NOAA) Physical Sciences Laboratory (PSL). Positive (exceeding +0.5 °C) and negative (lower than −0.5 °C) values of the Niño 3.4 SST index, respectively, indicate El Niño and La Niña conditions. The variability of the AAO is represented by the monthly AAO index from the NOAA/PSL, which is determined from Empirical Orthogonal Function analysis of monthly mean 700 hPa height anomalies. Note that an AAO index exceeding +0.5 and lower than −0.5 represent positive and negative phases of AAO, respectively.

### 2.3. Argo-Based Steric Sea Level Estimates

Monthly $1° \times 1°$ gridded data of temperature and salinity Argo profiles over 0–2000 dbar from the Roemmich-Gilson (RG) Climatology (updated from Roemmich and Gilson [45]) are used to examine the relative contributions of the steric components (SSL, TSL and HSL) to the interannual sea level variability in the South Pacific for the 2004–2020 period. Monthly estimates of SSL can be approximated as the sum of the TSL and HSL components:

$$\text{SSL} \approx \text{TSL} + \text{HSL} = -g^{-1} \left[ \int_H^h \delta_T dp + \int_H^h \delta_S dp \right] \tag{3}$$

where $\delta_T = \alpha(\overline{S}, T, p) - \alpha(35, 0, p)$ and $\delta_S = \alpha(S, \overline{T}, p) - \alpha(35, 0, p)$ are, respectively, the in-situ specific-volume ($\alpha$) anomalies (computed with the 1980 equation of state, EOS-80) for the thermosteric and halosteric components, $g$ is the gravity; $h$ and $H$ are the upper and lower pressure boundaries of the ocean layer, and $T$, $S$, $p$ are, respectively, the temperature, salinity and pressure. The overbars represent the mean temperature or salinity over time. Regional changes in SSL, TSL and HSL are obtained by removing the global mean from each component time series, as described earlier for SLA time series.

Recent studies [46,47] identified a significant and fast "salty drift" in the near-real time (NRT) Argo salinity profiles since about 2015. Consequently, a spurious decrease in global mean HSL has been observed with gridded Argo products, which predominantly rely on NRT salinity data after about 2015 rather than the delayed mode (DM) data [48]. Since the RG climatology quality control procedure [45] includes a step to correct salinity drifts by relaxing, when necessary, the NRT values towards the World Ocean Circulation Experiment (WOCE) climatology, the global mean HSL derived from RG climatology remains close to zero during 2015–2020 (Figure 1d in Barnoud et al. [48]; Figure S1). However, this drift correction applied to a large amount of salinity profiles (about 25% in August 2018) may bias the actual in-situ salinity changes beyond 2015 and may lead to an underestimation of recent freshening events.

In addition to RG Climatology, we also use the monthly $0.5° \times 0.5°$ Argo fields of the 8th version of the In Situ Analysis System (ISAS20; [49]; https://doi.org/10.17882/52367), because the ISAS products predominantly rely on DM data (i.e., with a lag of about 1 year after the NRT profiles). Starting from 2019, a decrease in global mean HSL derived from ISAS20 is observed (dashed blue line in Figure S1), reflecting the inclusion of drifted NRT salinity profiles due to the nonavailability of DM data. It is, however, unclear how the observed "salty drift" in Argo salinity data can affect HSL at regional scale. This question requires dedicated research and will need to be addressed in the near future; meanwhile, we acknowledge that the regional patterns of HSL change after 2015 presented in this study should be taken with caution.

### 2.4. 1.5 Layer Reduced-Gravity Model

The RGM model simulates the first-mode baroclinic midlatitude South Pacific sea level response to open-ocean wind-stress curl anomalies with respect to the 1993–2020

time mean (wind-stress forcing) and prescribed time-varying SLA signals at the eastern boundary of the model domain (eastern boundary forcing). Note that for the latter, the GMSL for the altimetry period of 1993–2020 was removed from SLA. We also consider here the contribution of the net surface heat flux (surface heat exchange) to represent a more complete picture of the surface forcing mechanisms involved. The interannual variations in SLA attributed to wind-forced and boundary-forced baroclinic processes are typically derived from the linear vorticity equation under the long-wave approximation (e.g., [50]) as:

$$\frac{\partial h_{br}}{\partial t} = c_R \frac{\partial h_{br}}{\partial x} - \frac{g\prime \nabla \times \tau\prime}{\rho_0 g f} - R h_{br} \tag{4}$$

where $\rho_0$ is the average density at the surface ocean, $h_{br}$ corresponds to SLA changes due to baroclinic processes, $c_R$ is the long baroclinic Rossby wave speed, $g$ and $g'$ are, respectively, the gravity and the reduced gravity, and $R$ is a frictional damping term. Sea level variations due to the net surface heat flux were modeled following Bowen et al. [39]:

$$\frac{\partial h_{sh}}{\partial t} = \frac{-\alpha Q_{net}}{\rho_0 C_p} \tag{5}$$

where $\alpha$ is the thermal expansion coefficient, $Q_{net}$ is the net surface heat flux (negative into the ocean), and $C_p$ is the specific heat capacity. Thus, the full model used in this study is:

$$\frac{\partial h_{br}}{\partial t} = c_R \frac{\partial h_{br}}{\partial x} - \frac{g\prime \nabla \times \tau\prime}{\rho_0 g f} - R h_{br} - \frac{\alpha Q_{net}}{\rho_0 C_p} \tag{6}$$

The zonal model domain (orange rectangle in Figure 1a) extends from $180°$ to the Chilean coast at $70°W$ (the eastern boundary). The model is initialized with SLA signals averaged over the $34°S$–$36°S$ band at the eastern boundary. The reduced gravity was determined as g′ = $c^2/H_p$, where the internal long gravity wave phase speed $c$ was set to 2.5 m s$^{-1}$ based on Chelton et al. [51], and a pycnocline depth $H_p$ of 140 m. A damping term of $R$ = 1.3 year$^{-1}$ and a long baroclinic Rossby wave speed $c_R$ of 0.05 m s$^{-1}$ were used as the optimal to obtain the best fit with observations for the $34°S$–$36°S$ averaged band.

### 3. Results and Discussion

To provide the first insight into the respective contributions of the steric components to the interannual sea level variability during 2004–2020, SLA (from satellite altimetry) and the upper-ocean SSL, TSL and HSL (from Argo) are spatially averaged over two boxes: (i) from $20°S$–$20°N$ and $110°W$–$180°$ (Figure 1b) in the tropical Pacific and (ii) from $30°$–$40°S$ and $130°W$–$180°$ in the southwest Pacific (Figure 1c). As expected, the time-varying SLA (black curves in Figure 1b,c) over 2004–2020 is largely due to TSL changes (orange curves in Figure 1b,c) in both boxes, while contributions from HSL changes (blue curves in Figure 1b,c) are limited to ±0.5 cm. From Figure 1b,c, we observe some detailed differences in amplitude between the changes in SLA and those of SSL. These differences can mostly be attributed to: (i) SSL changes in the deep-ocean, which cannot yet be directly considered here due to the lack of Argo data below 2000 dbar, (ii) removing global means based on different spatial coverage from SLA and the steric sea level components, and to a lesser extent, (iii) regional manometric (ocean mass-related) changes. Inherent errors in altimetric measurements and Argo data might also explain part of the observed differences. Overall, we also note a good agreement between the steric components derived from the RG Climatology (solid curves in Figure 1b,c) and ISAS20 (dashed curves in Figure 1b,c), although some differences (notably in TSL) are observed during 2014–2016. This suggests that using Argo-based steric data either from the RG climatology or ISAS20 can be considered for assessing the respective contributions of TSL and HSL (despite the GMHSL drift seen with ISAS20 in 2019 and beyond) to the interannual sea level variability in the tropical Pacific and the midlatitude southwest Pacific.

As mentioned earlier, the interannual sea level variability in the tropical Pacific is largely influenced by ENSO dynamics and associated (horizontal and vertical) redistri-

bution of heat (e.g., [19]). This is further illustrated in Figure 1b, where both TSL time series averaged over 20°S–20°N and 110°W–180° are strongly correlated (at zero-time lag) with the 1.5-year low-pass filtered Niño 3.4 index ($r$ = 0.84 with a 95% significance level for correlation estimated at 0.52). In the southwest Pacific box (30°–40°S and 180°–130°W; Figure 1c), the interannual variations of SLA and TSL from 2006 through 2013 depict the wind-forced ocean heat buildup that occurred during the 2005–2014 decade, characterized by persistent La Niña-like conditions (e.g., [28–30]). However, contrary to the tropics and despite a decrease in sea level by 3 cm in both SLA and TSL during the 2015–2016 El Niño, the correlation between SLA (or TSL) and the 1.5-year low-pass filtered Niño 3.4 SST index is not significant (at 95% confidence level) for the 2004–2020 period. The correlation is also not significant between SLA (or TSL) and the 1.5-year low-pass filtered AAO index. In line with past studies (e.g., [34,52–54]), these findings suggest that interannual sea level changes since 2004 in the southwest Pacific region from 30°S to 40°S are rather the result of a large-scale oceanic adjustment (with some time lag) to a complex interplay between the low-frequency modulation of ENSO and the AAO in addition to intrinsic ocean variability.

In the following two sections, we will focus on the 2015–2020 period, which marks the end of the nearly decade-long increase in sea level in the South Pacific as El Niño started to develop in 2015.

### 3.1. Basin-Scale Sea Level Changes and Local Wind Forcing in 2015–2020

During the strong 2015–2016 El Niño, the western and eastern tropical Pacific were associated with a decrease and increase in sea level, respectively, as shown in Figure 2a by the SLA differences between 2016 and 2014. This is due to the eastward and meridional redistributions of ocean HC across the tropical Pacific and from 20°S–5°N into the North Pacific region of 5°N–20°N, respectively, as typically observed during eastern Pacific El Niño events (e.g., [19,55,56]). The SLA increase and associated warming south of about 25°S in the Tasman Sea (Figure 2a) is rather well explained by the Ekman solution $\Delta h_{Ek}$ (color shading in Figure 2b) given by Equation (2). This ocean warming (outlined by the red shading in Figure 2b and Figure S2b) was driven by anticyclonic surface wind anomalies in 2015–2016 (arrows in Figure 2b and Figure S2b) via downward Ekman pumping anomalies south of 30°S between 150°E–180° (Figure S2b). In contrast, the Ekman solution $\Delta h_{Ek}$ does not fully explain the observed SLA changes south of 20°S and east of New Zealand, although anomalous cyclonic winds (Figure 2b and Figure S2b) and associated upward Ekman pumping anomalies between 130°W–160°W (Figure S2b) could have contributed to the SLA decrease observed in this region (Figure 2a). This is probably because the westward propagation of Rossby waves plays an important role in the SLA variability at midlatitudes in the South Pacific (e.g., [31,39]).

In Figure 2c, opposite SLA differences are observed between 2018 and 2016 in the tropical Pacific, reflecting the ocean heat recharge in the tropical western Pacific during the 2017–2018 La Niña as a response to intensified trade winds in the equatorial region. South of 20°S, SLA decreased for the most part in the Tasman Sea (Figure 2c), however, $\Delta h_{Ek}$ (Figure 2d) can only explain the SLA decrease in a rather small area of the eastern Tasman Sea close to New Zealand, possibly due to anomalous cyclonic winds (Figures 2d and S2d) and associated anomalous Ekman pumping (Figure S2d) in 2017–2018. East of New Zealand, SLA increased south of 30°S between 130°W–180° and in the region south of 20°S between 90°–120°W (Figure 2c). These local increases in SLA are rather well captured by $\Delta h_{Ek}$ (Figure 2d), suggesting that the local anticyclonic wind-related Ekman pumping anomalies (Figure S2d) contributed at least in part to the increasing patterns of SLA.

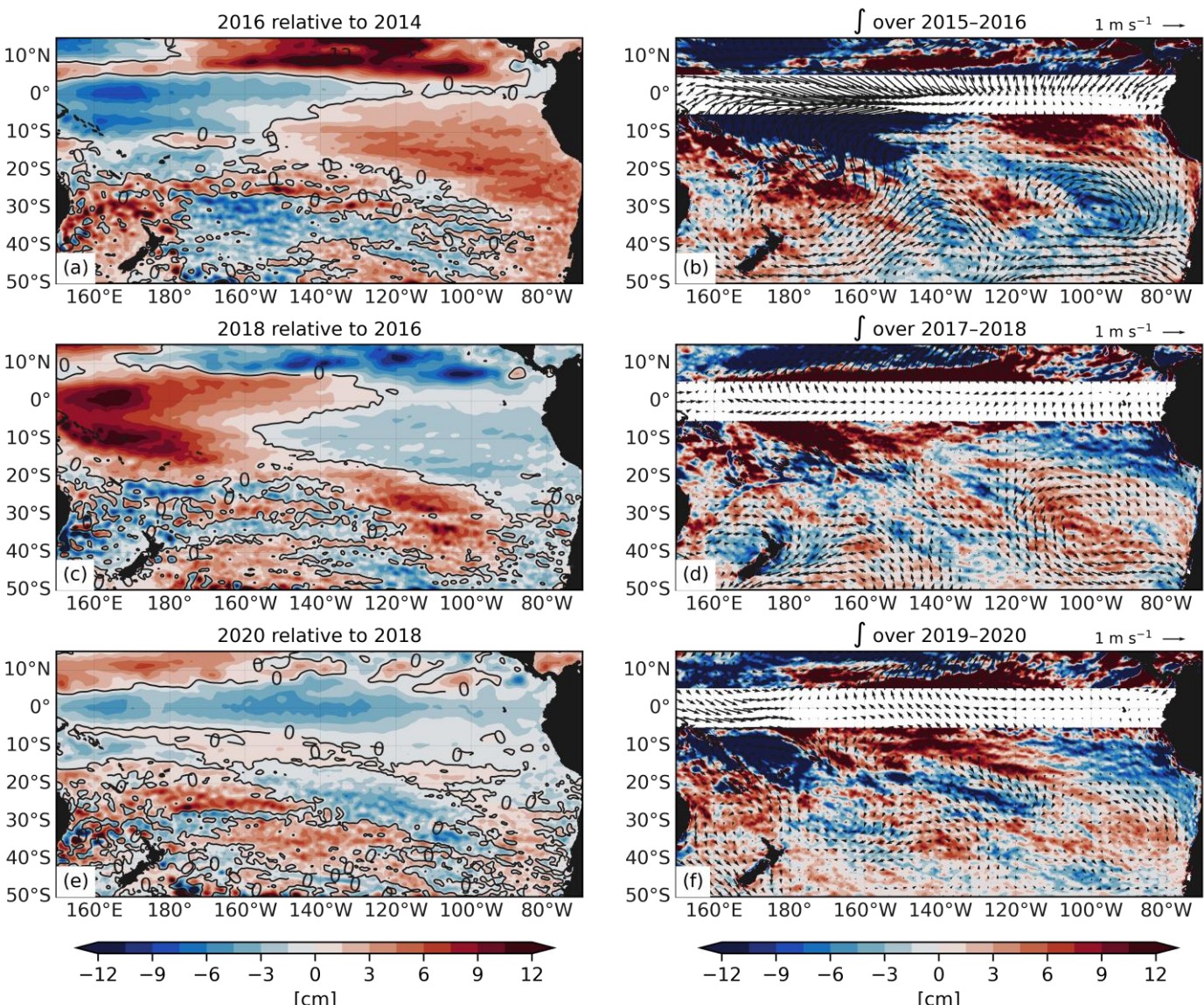

**Figure 2.** Differences in sea level anomaly (SLA) derived from satellite altimetry: (**a**) 2016 minus 2014, (**c**) 2018 minus 2016 and (**e**) 2020 minus 2018. The Ekman solution $\Delta h_{Ek}$ (color; see Equation (2)) is computed by integrating over time the Ekman pumping anomalies $W_{Ek}$ relative to the 1993–2020 time mean over (**b**) 2015–2016, (**d**) 2017–2018 and (**f**) 2019–2020. The 5°S–5°N region is excluded from the computation because the Coriolis parameter tends towards zero near the equator, thereby Ekman pumping cannot be properly determined. The 10-m wind velocity anomalies (relative to the 1993–2020 time mean) averaged over (**b**) 2015–2016, (**d**) 2017–2018 and (**f**) 2019–2020 are shown by the arrows.

The SLA differences between 2020 and 2018 (Figure 2e) coincided with ENSO-neutral conditions. In the tropical Pacific, SLA increased primarily in the western part of the basin north of 5°N, but it decreased within the 20°S–5°N band across the basin (Figure 2e). While SLA primary increased west and east of New Zealand south of 20°S (Figure 2e), the Ekman solution (Figure 2f) cannot well reproduce the observed SLA patterns except in the region of 130°W–140°W and at about 80°W, where anomalous anticyclonic winds (Figures 2f and S2f) and related downward Ekman pumping anomalies (Figure S2f) are found in 2019–2020. This further suggests that interannual SLA changes in the midlatitude South Pacific are predominantly driven by a combination of local and remote forcing mechanisms, which needs to be better clarified. In this regard, the respective contributions of the wind and

eastern boundary forcings under Rossby wave characteristics to the interannual sea level variability will be determined for the 2004–2020 period in Section 3.3.

During the 2015–2016 El Niño, we observe negative westerly wind anomalies south of 35°S and east of 120°W (Figure 2b), while positive anomalies are found during the 2017–2018 La Niña (Figure 2d). Both these opposite westerly wind anomalies coincided with a positive phase of the 1.5-year low-pass filtered AAO index (gray shading in Figure 1c). This is surprising, since previous studies identified that weakened westerly winds often occurred during positive AAO phases, whereas intensified westerly winds often occurred during negative AAO phases (e.g., [21]). Note that in contrast to the 2015–2016 El Niño and the 2017–2018 La Niña, there was no weakening or strengthening of the westerly winds south of 35°S in the eastern part of the basin (75°W–120°W) during 2019–2020 (Figure 2f), which is a period characterized by a neutral phase of the low-pass filtered AAO index (Figure 1c). Previous studies [57,58] also identified that a negative AAO phase often coincides with El Niño conditions, whereas a positive AAO phase often coincides with La Niña conditions, during austral summer. However, this relationship between ENSO and the AAO in the South Pacific warrants further investigation, notably over the past decade (since about 2010), which is dominated by positive AAO phases (Figure 1c). This necessary effort is, however, beyond the scope of this study and is thus left for future research.

Furthermore, the local changes in wind forcing during the 2015–2016 El Niño and the 2017–2018 La Niña led to both lateral and zonal redistributions of HC in the upper ocean layers which in turn may have significantly contributed to the observed sea level changes during 2015–2020. This is examined in the section below, in addition to contributions from freshwater content changes.

### 3.2. Basin-Scale Thermosteric and Halosteric Sea Level Changes in 2015–2020

The upper 2000 dbar TSL differences between 2016 and 2014 (Figures 3a and S3a) are, as expected, in a good agreement with the corresponding SLA differences (Figure 2a). The observed TSL changes occurred primarily in the 100–500 dbar layer (Figures 3b and S3b), reflecting the large redistribution of HC associated with the 2015–2016 El Niño. Significant contributions from the TSL changes over 0–100 dbar are also observed in the central/eastern tropical Pacific and the western equatorial region (Figures 3c and S3c), as a response to horizontal and vertical heat advection, and changes in net surface heat flux (e.g., [56]). South of 20°S, most of the SLA changes observed west and east of New Zealand (Figure 2a) can be attributed to TSL changes within the 100–500 dbar layer. Significant TSL changes also occurred in the 500–2000 dbar deeper layer (Figures 3d and S3d), notably west of 160°E, in the western boundary region from 30°S to 40°S. The corresponding HSL differences over the 0–2000 dbar (Figures 4a and S4a), 100–500 dbar (Figures 4b and S4b), 0–100 dbar (Figures 4c and S4c) and 500–2000 dbar (Figures 4d and S4d) layers show that HSL changes can partially compensate for the TSL changes in the tropical Pacific and the western boundary region of 30°S–40°S. In the western tropics, HSL mostly increased over 0–100 dbar (Figures 4c and S4c) as a result of a large freshening event caused by a westward redistribution of freshwater content and changes in buoyancy forcing at the onset of the 2015–2016 El Niño (e.g., [59]). In the western boundary region, HSL decreased over 100–500 dbar (Figures 4b and S4b), possibly because of salty anomalies linked to the EAC transport variability and related anticyclonic eddies ([60]; their Figure 2).

The differences in TSL over 0–2000 dbar between 2018 and 2016 (Figures 5a and S5a) reflect the redistribution of the upper ocean HC across the basin during the 2017–2018 La Niña (Figure 2c). In the western and eastern tropical Pacific, TSL increased and decreased, respectively, in the 100–500 dbar layer (Figures 5b and S5b), featuring the typical recharge of the "warm pool" via a westward redistribution of heat (e.g., [56]). TSL over 0–100 dbar (Figures 5c and S5c) decreased in the eastern tropics (away from the 5°S–5°N region), as a result of both adiabatic and diabatic redistributions of heat related to the 2017–2018 La Niña (e.g., [10]). South of 20°S, the differences in TSL exhibit opposite spatial patterns than those during the 2015–2016 El Niño, with TSL changes occurring primarily within

the 100–500 and 500–2000 dbar layers (Figures 5d and S5d). The HSL differences between 2018 and 2016 over 0–2000 dbar (Figures 6a and S6a) further show that HSL changes can dampen the changes in TSL observed in the region. Sizable contributions from the HSL changes occurred over 100–500 dbar (Figures 6b and S6b) and 0–100 dbar (Figures 6c and S6c), while the changes over 500–2000 dbar (Figures 6d and S6d) played only a minor role.

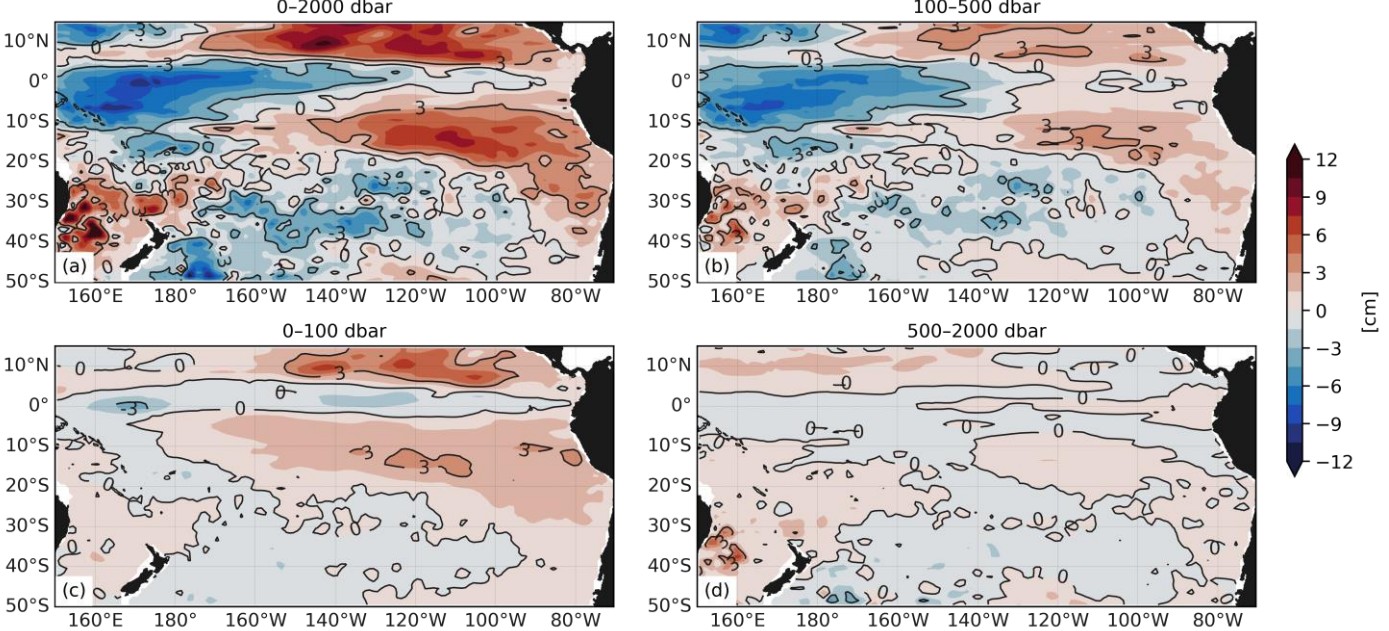

**Figure 3.** Differences in the Roemmich-Gilson Argo-based thermosteric sea level (TSL) for 2016 minus 2014 over (**a**) 0–2000 dbar, (**b**) 100–500 dbar, (**c**) 0–100 dbar and (**d**) 500–2000 dbar. Contour lines (see inline numbers for TSL intervals) are also shown.

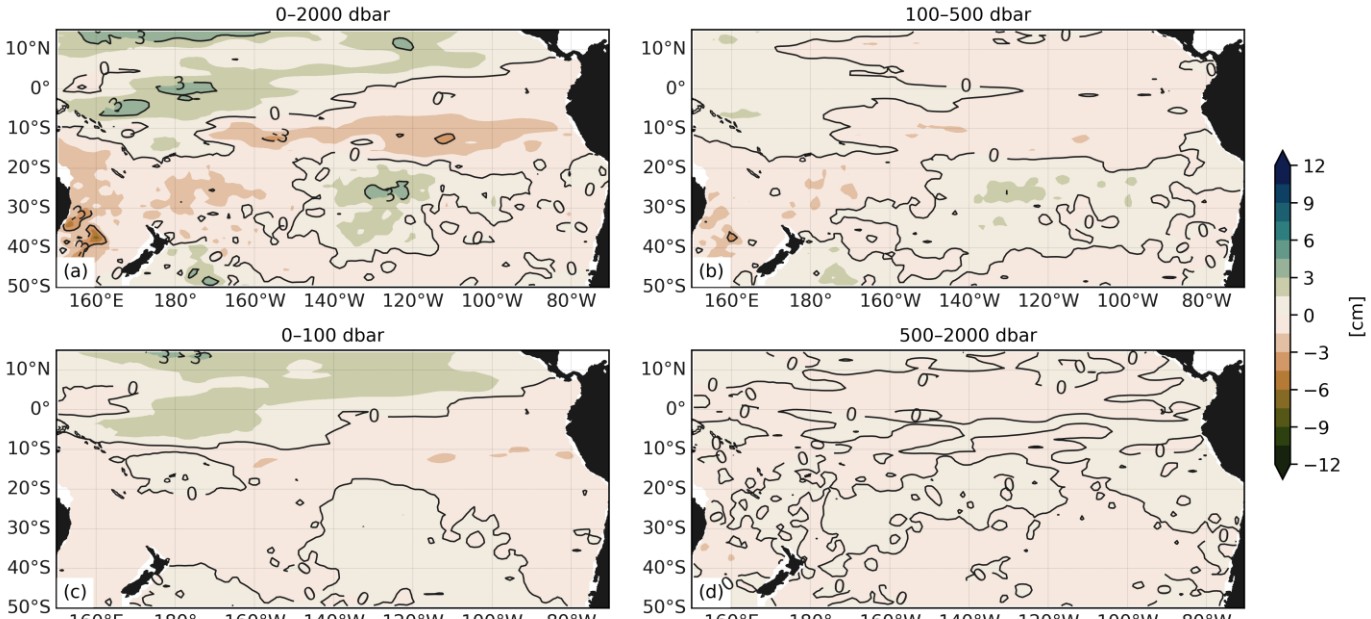

**Figure 4.** Differences in the Roemmich-Gilson Argo-based halosteric sea level (HSL) for 2016 minus 2014 over (**a**) 0–2000 dbar, (**b**) 100–500 dbar, (**c**) 0–100 dbar and (**d**) 500–2000 dbar. Contour lines (see inline numbers for HSL intervals) are also shown.

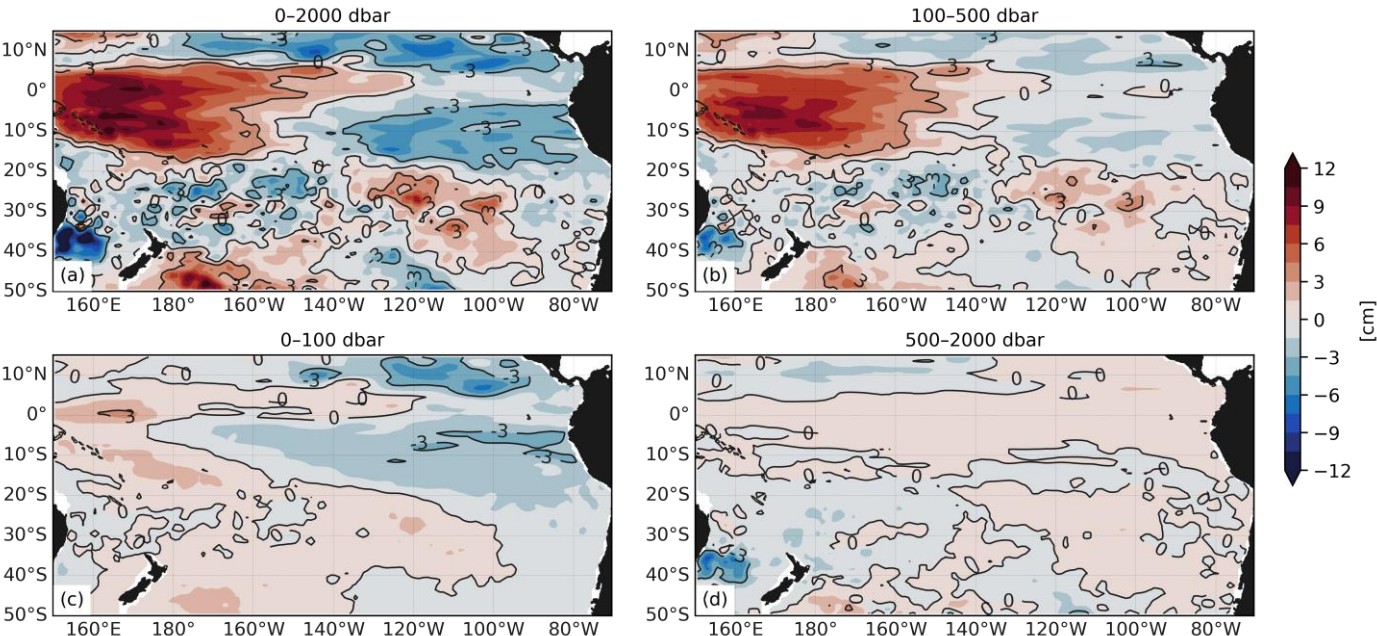

**Figure 5.** Differences in the Roemmich-Gilson Argo-based thermosteric sea level (TSL) for 2018 minus 2016 over (**a**) 0–2000 dbar, (**b**) 100–500 dbar, (**c**) 0–100 dbar and (**d**) 500–2000 dbar. Contour lines (see inline numbers for TSL intervals) are also shown.

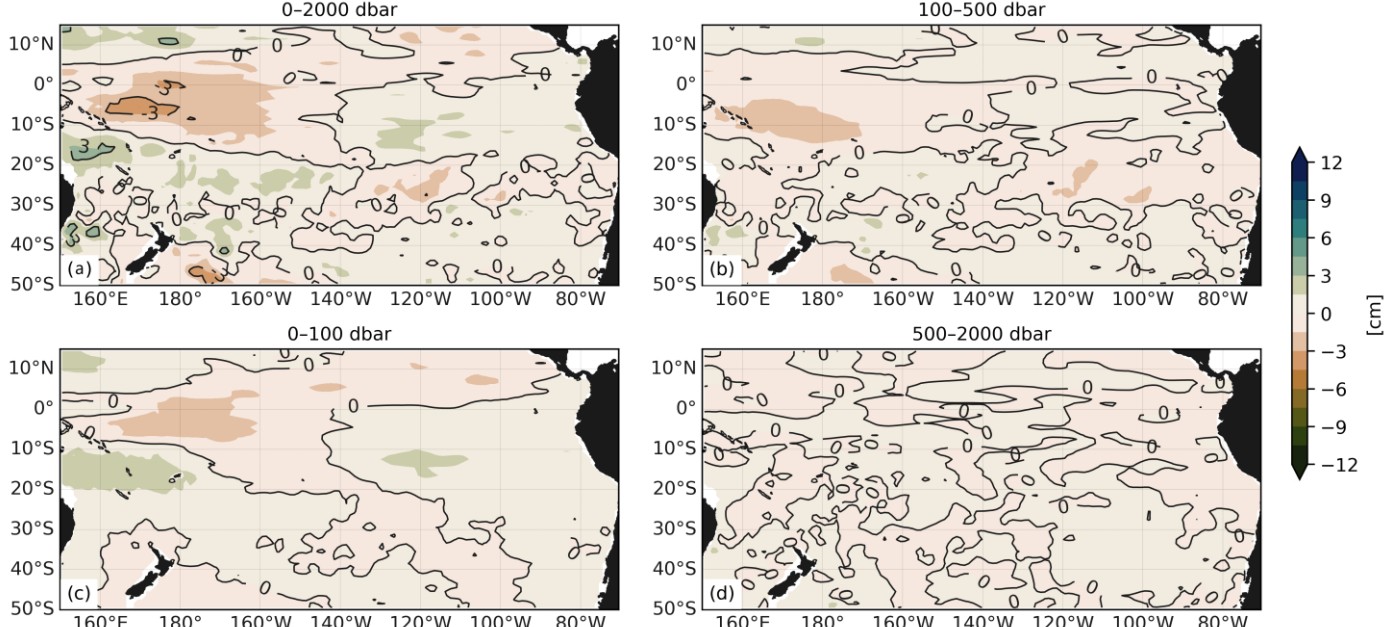

**Figure 6.** Differences in the Roemmich-Gilson Argo-based halosteric sea level (HSL) for 2018 minus 2016 over (**a**) 0–2000 dbar, (**b**) 100–500 dbar, (**c**) 0–100 dbar and (**d**) 500–2000 dbar. Contour lines (see inline numbers for HSL intervals) are also shown.

The SLA differences between 2020 and 2018 shown in Figure 2e agree well with the corresponding differences in TSL over 0–2000 dbar (Figures 7a and S7a). Most of the changes occurred primarily over 100–500 dbar (Figures 7b and S7b), where heat is redistributed from the equatorial region into the tropics (within 20°–5°S and poleward of 5°N). Significant changes also occurred at 30°–35°S across the basin. Over 0–100 dbar, TSL decreased in the central and eastern tropical Pacific (Figures 7c and S7c), likely due to changes in net surface heat fluxes. Over 500–2000 dbar, some sizable contributions from the

TSL changes are observed in the western boundary region of 30°–40°S (Figures 7d and S7d), due to wind-driven isopycnal heaving and/or variations in heat transport by the poleward EAC. The HSL differences between 2020 and 2018 (not shown) are overall within ±1 cm, suggesting that HSL changes did not significantly contribute to regional sea level change from 2018 to 2020.

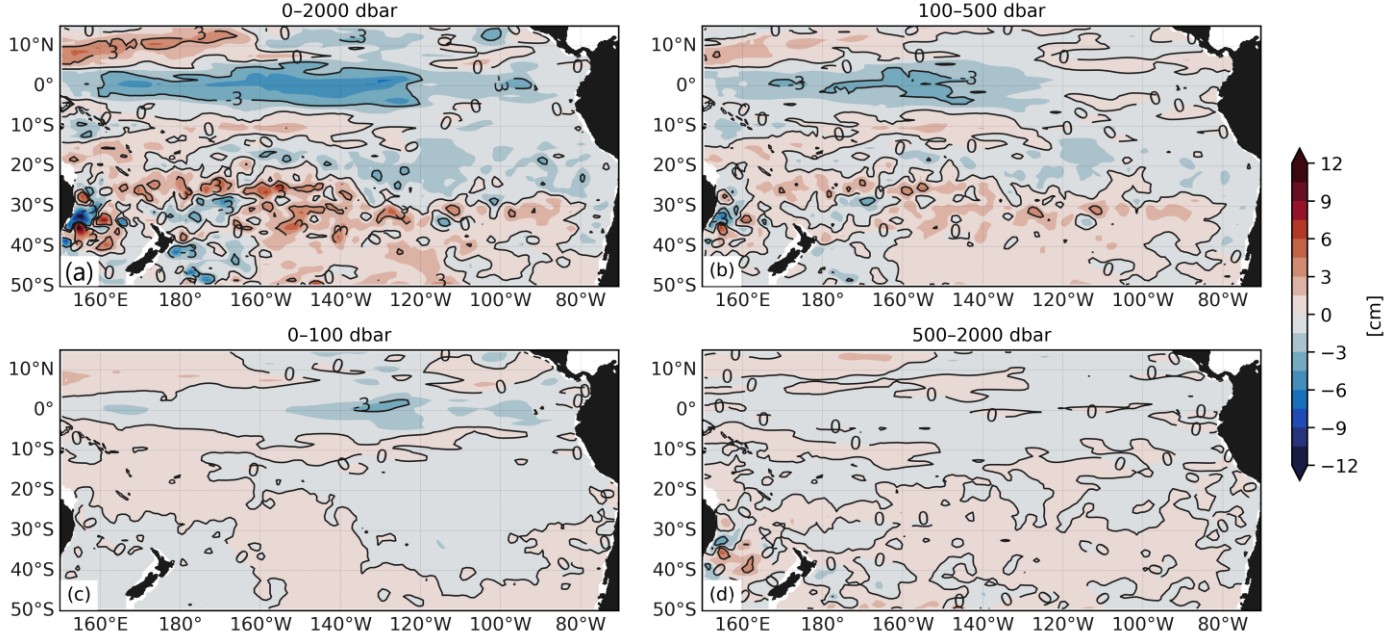

**Figure 7.** Differences in the Roemmich-Gilson Argo-based thermosteric sea level (TSL) for 2020 minus 2018 over (**a**) 0–2000 dbar, (**b**) 100–500 dbar, (**c**) 0–100 dbar and (**d**) 500–2000 dbar. Contour lines (see inline numbers for TSL intervals) are also shown.

### 3.3. Wind-Driven Rossby Waves and Surface Heat Flux Forcing

In the eastern South Pacific, changes in local Ekman pumping at midlatitudes have been shown to trigger westward propagating SLA signals in 2007 and over 2010–2013 (see [29]), reaching New Zealand in about 7–8 years. The time-longitude diagrams of SLA (Figure 8a) and TSL (Figure 8b) averaged between 34°–36°S during the 2004–2020 period further indicate that SLA signals (dominated by TSL changes) propagated westward as Rossby waves across the basin from the 100°–120°W region, where large anomalies in Ekman pumping (Figure 8c) are frequently observed. To gain more insight into the contribution of Rossby waves induced by the open-ocean wind-stress forcing and SLA changes at the eastern boundary (in addition to the effects of changes in net surface heat fluxes), the RGM results averaged between 34°–36°S (hereinafter referred to as 35°S) for the 2004–2020 period are examined below.

The observed time-varying SLA at 35°S in 2004–2020 (Figure 9a) are reasonably well reproduced by the RGM (Figure 9b), although some differences exist (e.g., the positive SLA pattern over the 160°W–180° region in 2006 and the negative propagating SLAs west of about 120°W from 2011–2014 in the model). The alternating spatial patterns of positive and negative SLA seen in the observations (notably west of 100°W) are well captured by the model. In Figure 9c, the RGM is only forced with wind-stress curl anomalies. This model solution represents the observed spatial patterns of SLA very well, notably west of 100°W, confirming that wind-forced baroclinic Rossby waves can largely contribute to the interannual sea level variability in the South Pacific at 35°S. The model solution related to the eastern boundary forcing alone (Figure 9d) can well simulate the observed westward propagating SLA signals in the region east of about 100°W, but it is much less successful in simulating the SLA variations in the open-ocean region of the midlatitude South Pacific. It

is also interesting to note that similar conclusions can be drawn for the period since the start of the altimetry era, from 1993 to 2020 (Figure S8).

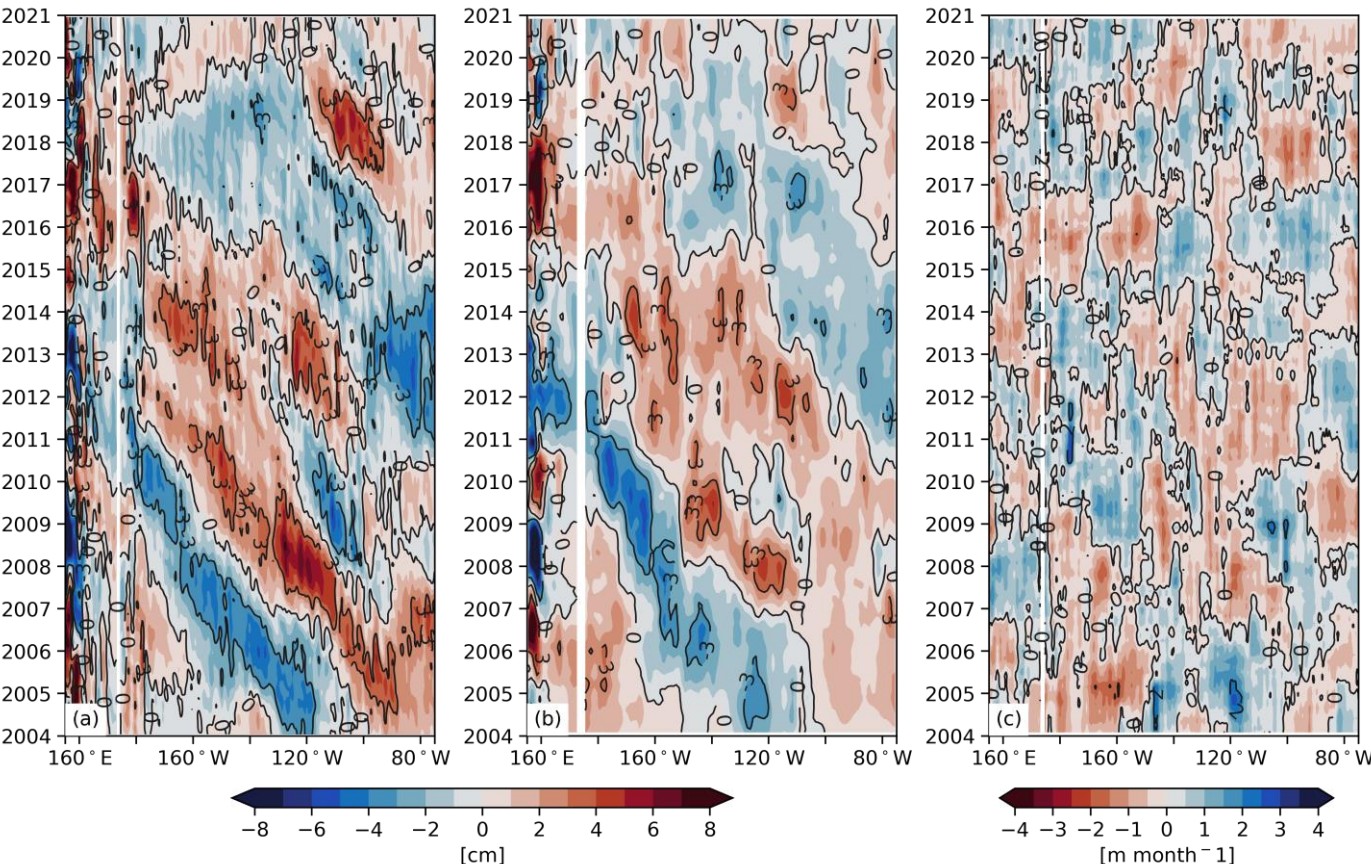

**Figure 8.** (**a**) Time-Longitude diagrams from 155°E to 75°W at 35°S (by averaging over 34°–36°S) in the South Pacific during 2004–2020 of (**a**) observed interannual sea level anomaly derived from altimetry, (**b**) Roemmich-Gilson Argo-based interannual thermosteric sea level and (**c**) interannual Ekman pumping anomaly (positive upward). To emphasize on local changes along 35°S, the zonal mean was removed from all-time series. Contour lines (see inline numbers for intervals) are also shown.

In Figure 9e,f, we also compare the observed and modeled patterns of SLA (forced by each surface forcing term) averaged in the 100°W–180° and 75°–100°W regions, respectively. As expected from the findings presented above, the contributions from the wind-stress forcing (red curve in Figure 9e) can largely explain the temporal SLA changes (black curve in Figure 9e) seen in the region west of 100°W. However, both the wind-stress forcing (red curve in Figure 9f) and the eastern boundary forcing (blue curve in Figure 9f) contributed to the observed time-varying SLA in the region east of 100°W. The impact of the surface heating forcing (green curves in Figure 9e,f) is found to be negligible in both regions.

While a significant fraction (40 up to 75%) of the variance in SLA west of 100°W is explained by the wind-stress forcing during the 2004–2020 period (solid red line in Figure 10a), one should note that this fraction of SLA variance is, however, smaller for the period of 1993–2020, as it explains only up to 60% the variance (dotted red line in Figure 10a). This finding suggests that the wind-stress forcing contributions have become even more dominant since 2004. The eastern boundary forcing explains 20 up to 40% of the variance in the 80°–100°W band during 2004–2020 (see solid blue line in Figure 10a), and up to 50–60% for the longer period of 1993–2020 (dotted blue line in Figure 10a). Interestingly, note that forcing the RGM only with wind-stress curl anomalies in the regions of 100°–140°W (blue line in Figure 10b) and 100°–120°W (green line in Figure 10b), confirms that the local wind-stress curl anomalies generated in the latter region are important to remotely force the

propagating SLA signals west of 100°W up to about 140°W. It also confirms that the SLA variations west of 165°W–170°W are primarily forced by local wind forcing (Figure 10b).

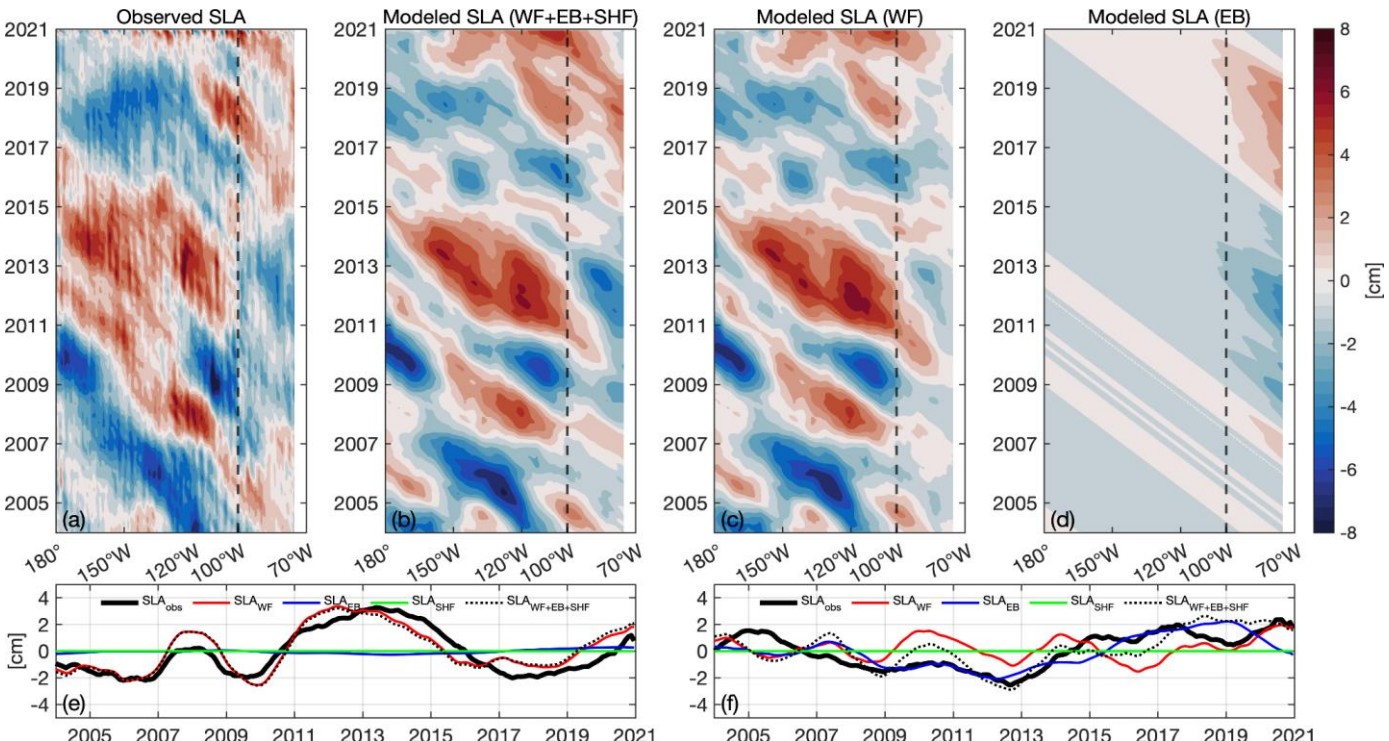

**Figure 9.** (**a**–**d**) Time-Longitude diagrams of mid-South Pacific (averaged over 34°S–36°S) interannual sea level anomaly (SLA) signals from 180° (east of New Zealand) to 75°W (at the eastern boundary, close to the Chilean coast) during 2004–2020. (**a**) Observed SLA from satellite altimetry (the 1993–2020 global mean sea level is removed). (**b**) Modeled SLA due to wind-forced baroclinic Rossby waves, time-varying SLA signals at the eastern boundary and diabatic surface heating. (**c**) Modeled SLA patterns forced only by wind-stress curl anomalies (WF), ignoring SLA signals propagating from the eastern boundary. (**d**) Modeled SLA considering the eastern boundary forcing (EB) only. (**e**,**f**) Time series of (i) the observed SLA (solid black curves), (ii) modeled SLA with wind forcing only (red curves), (iii) modeled SLA with the eastern boundary forcing only (blue curves), (iv) modeled SLA with the net surface heat flux forcing (derived from ERA5 atmospheric reanalysis) only (green curves), and (v) modeled SLA combining all forcing (dotted black curves) averaged over 100°W–180° in (**e**) and 75°W–100°W in (**f**).

Despite large uncertainties in the net surface heat flux estimates from atmospheric reanalysis (e.g., [27]), we found that the contributions from the net surface heating forcing to interannual sea level change are negligible. Indeed, the sea level changes forced by variable surface heat fluxes from ERA5 (Figure S9a), NCEP1 (Figure S9b) and JRA55 (Figure S9c) during 2004–2020 are comparable, but approximately an order of magnitude smaller than the response to wind-stress forcing.

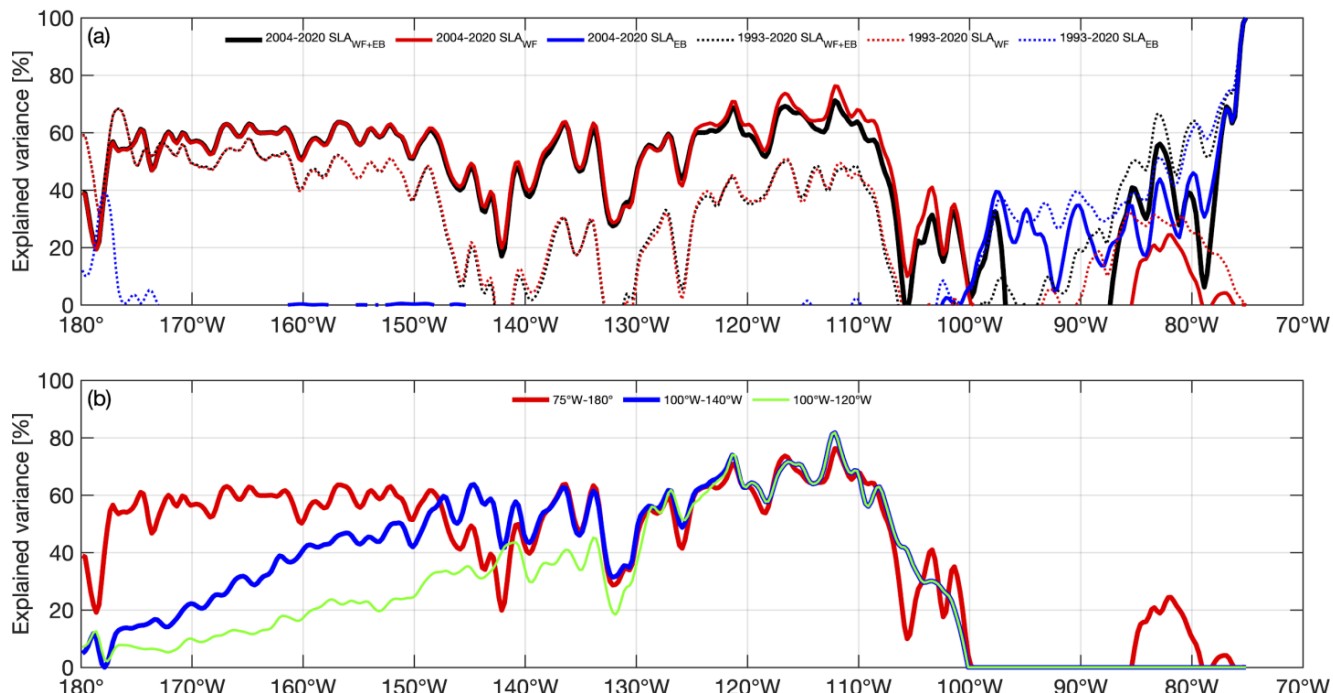

**Figure 10.** (**a**) Fraction of variance (%) of the South Pacific interannual sea level anomaly (SLA) explained by the 1.5 layer reduced-gravity model from 180° to 75°W and averaged between 34°S and 36°S during 2004–2020 (solid lines) and for the period of 1993–2020 (dotted lines). The fraction of variance attributed to the combined effects of the wind-stress and eastern boundary forcing terms is in black; the fraction of variance due to the wind-stress forcing only is in red, and that of the boundary forcing only is in blue. (**b**) Fraction of SLA variance (%) attributed to the wind-stress forcing in the 180°–75°W band (red line; as in (**a**)), the 140°W–100°W band (blue line) and the 120°W–100°W band (green line) for the period of 2004–2020.

## 4. Conclusions

During the 2005–2014 decade, the ocean HC increased in the subtropical southwest Pacific as a response to persistent patterns of wind-stress anomalies and related Ekman convergence during La Niña dominant conditions (2007–2009, 2010–2012) (e.g., [29,30]), leading to a significant sea level increase in the midlatitude southwest Pacific. In this study, we further investigated the surface forcing mechanisms of sea level change at interannual time scales in the midlatitude South Pacific (centered at 35°S) during 2004–2020, with an emphasis on the 2015–2020 period, which has been marked by the strong El Niño of 2015–2016 and the La Niña event of 2017–2018.

During the 2015–2016 El Niño, an increase in TSL (mainly over 100–500 dbar, but also within the 500–2000 dbar layer) occurred in the vicinity of the western boundary from 30°S to 40°S, west of New Zealand. This increase is shown to be partially compensated by a decrease in HSL within the 100–500 dbar layer, due to the vertical deflection of the thermocline and associated salty anomalies. Opposite wind forcing during the 2017–2018 La Niña yielded reversed spatial patterns in TSL and HSL. Below 2000 dbar, the lack of measurements on year-to-year time scales still prevents us from investigating the deep-ocean (over 2000–4000 dbar) and abyssal (over 4000–6000 dbar) contributions to sea level changes, notably in the region west of New Zealand. By means of Deep-Argo floats, a warming over 5000–5600 dbar from mid-2014 to 2018 has been, however, reported by Johnson et al. [61] in the Southwest Pacific Basin, east of New Zealand (see Figure 1a). This warming suggests a rather large deep-ocean contribution of about 1 mm yr$^{-1}$ to steric sea level change in the Southwest Pacific Basin over that period. One may thus expect that the ongoing extension of the Deep-Argo array to the entire South Pacific Ocean (and ultimately

the world's oceans) will greatly improve our ability to quantify the relative contribution of the deep and abyssal oceans to sea level variability on seasonal-to-decadal time scales.

We used a 1.5-layer reduced-gravity model to simulate interannual sea level patterns averaged between 34°S–36°S across the South Pacific basin from 180° to 75°W. The model was forced by (i) changes in open-ocean surface wind-stress curl, (ii) time-varying SLA signals at the eastern boundary of the basin, and (iii) changes in net surface heat fluxes. In line with past studies, the model's results suggest that the interannual sea level variability in the midlatitude South Pacific is primarily due to changes in the wind-stress forcing. Indeed, we found that the wind-forced SLA signals explained, overall, more than 40% of the SLA variance west of 100°W (and up to 75% around 110°W) during 2004–2020. East of 100°W, a significant fraction of the variance can be attributed to SLA signals emanated from the eastern boundary, however, the contribution of local wind-stress forcing between the eastern boundary and 100°W remained important. Additionally, we found that the fraction of SLA variance explained by the wind-stress forcing is smaller (only up to 50–60%) during 1993–2020 than during 2004–2020. In this study, we also accounted for the contribution of surface heat fluxes to the observed interannual variability of SLA, which turned out to be negligible. This result somewhat disagrees with Bowen et al. [39], who suggested that surface heat fluxes could have explained a significant fraction of the SLA variance at 24°S during 1993–2004 on interannual time scales. However, it is consistent with Llovel and Terray [27], who found a negligible impact of variable surface heat fluxes on the interannual variability in ocean HC and TSL during the 2005–2014 decade in the southwest Pacific region.

The simple, although realistic, modeling framework of regional sea level change presented here explains a large part, but not all, of the interannual SLA variability in the region. The remaining part of the variability might largely be due to chaotic intrinsic ocean changes, particularly in the region west of New Zealand, where the intrinsic fluctuations of the EAC (and its eastern/southern extensions) have been shown to explain 40 to 60% of the interannual variability in regional sea level [12].

At last, this study advocates for a better understanding of the covariability between ENSO and the AAO wind forcing terms, especially since the past decade. This research effort provides useful insights for future analyses of interannual-to-decadal sea level changes in the region and their potential impacts on the South Pacific island communities.

**Supplementary Materials:** The following supporting information can be downloaded at: https://www.mdpi.com/article/10.3390/rs15020352/s1, Figure S1: Time series of the global mean thermosteric sea level (GMTSL; orange curves) and halosteric sea level (GMHSL; blue curves) anomalies relative to the 2006–2015 mean (as in Barnoud et al., 2021) for the 2004–2020 period and derived from two Argo-based products: (i) the Roemmich-Gilson (RG) Climatology (solid curves) and (ii) the 8th version of the In Situ Analysis System, ISAS20 (dashed curves). For each time series, the monthly mean climatology was removed, and a 3-month low-pass filter was applied to remove remaining subseasonal signals. The dominant influence of the Argo "salty drift" (https://argo.ucsd.edu/data/data-faq/#sbepsal; last access on 2 January 2023) on ISAS20 salinity data is seen in 2019 (dashed vertical line) and onwards due to the lack of delayed-mode adjusted salinity profiles; Figure S2: As in Figure 2, differences in sea level anomaly (SLA) derived from satellite altimetry for 2016 minus 2014 in (**a**) 2018 minus 2016 in (**c**) and 2020 minus 2018 in (**e**). Ekman pumping anomalies $W_{Ek}$ (color; positive upward) with respect to the 1993–2020 time mean and averaged over (**b**) 2015–2016, (**d**) 2017–2018 and (**f**) 2019–2020 instead of the Ekman solution shown in Figure 2. $W_{Ek}$ values over 5°S–5°N are excluded (as the Coriolis parameter tends towards zero near the equator) and the corresponding 10-m wind velocity anomalies are also shown (arrows); Figure S3: Differences in the 8th version of the In Situ Analysis System (ISAS20) Argo-based thermosteric sea level (TSL) for 2016 minus 2014 over (**a**) 0–2000 dbar, (**b**) 100–500 dbar, (**c**) 0–100 dbar and (**d**) 500–2000 dbar. Contour lines (see inline numbers for TSL intervals) are also shown. In each panel, white shading indicates areas where bathymetry is shallower than each pressure reference level; Figure S4: Differences in the 8th version of the In Situ Analysis System (ISAS20) Argo-based halosteric sea level (HSL) for 2016 minus 2014 over (**a**) 0–2000 dbar, (**b**) 100–500 dbar, (**c**) 0–100 dbar and (**d**) 500–2000 dbar. Contour

lines (see inline numbers for TSL intervals) are also shown. In each panel, white shading indicates areas where bathymetry is shallower than each pressure reference level; Figure S5: Differences in the 8th version of the In Situ Analysis System (ISAS20) Argo-based thermosteric sea level (TSL) for 2018 minus 2016 over (**a**) 0–2000 dbar, (**b**) 100–500 dbar, (**c**) 0–100 dbar and (**d**) 500–2000 dbar. Contour lines (see inline numbers for TSL intervals) are also shown. In each panel, white shading indicates areas where bathymetry is shallower than each pressure reference level; Figure S6: Differences in the 8th version of the In Situ Analysis System (ISAS20) Argo-based halosteric sea level (HSL) for 2018 minus 2016 over (**a**) 0–2000 dbar, (**b**) 100–500 dbar, (**c**) 0–100 dbar and (**d**) 500–2000 dbar. Contour lines (see inline numbers for TSL intervals) are also shown. In each panel, white shading indicates areas where bathymetry is shallower than each pressure reference level; Figure S7: Differences in the 8th version of the In Situ Analysis System (ISAS20) Argo-based thermosteric sea level (TSL) for 2020 minus 2018 over (**a**) 0–2000 dbar, (**b**) 100–500 dbar, (**c**) 0–100 dbar and (**d**) 500–2000 dbar. Contour lines (see inline numbers for TSL intervals) are also shown. In each panel, white shading indicates areas where bathymetry is shallower than each pressure reference level; Figure S8: Same as Figure 9, but for the period starting from the beginning of the satellite altimetry era, 1993 to 2020; Figure S9: (**a**–**c**) Time-Longitude diagrams of interannual sea level anomaly (averaged over 34°–36°S from 180° to 75°W) during 2004–2020 due to net surface heat forcing (SHF) only. Modeled SLA considering the SHF derived from (**a**) ERA5 (**b**) NCEP1 and (**c**) JRA55.

**Author Contributions:** Conceptualization, C.G. and D.L.V.; methodology, C.G., D.L.V., S.C. and W.L.; software, C.G. and D.L.V.; validation, C.G., D.L.V., S.C. and W.L.; formal analysis, C.G.; investigation, C.G.; resources, C.G., D.L.V. and S.C.; data curation, C.G.; writing—original draft preparation, C.G.; writing—review and editing, C.G., D.L.V., S.C. and W.L.; visualization, C.G.; supervision, D.L.V., S.C. and W.L.; project administration, D.L.V., C.G. and S.C.; funding acquisition, D.L.V. All authors have read and agreed to the published version of the manuscript.

**Funding:** This research was funded by the NASA Ocean Surface Topography Science Team program grant number NNX17AH59G) and the base funds of the NOAA Atlantic Oceanographic and Meteorological Laboratory. Cyril Germineaud also acknowledges support from the Cooperative Institute for Marine and Atmospheric Studies (CIMAS), a Cooperative Institute of the University of Miami and NOAA (cooperative agreement NA20OAR4320472), and from the CNRS and its joint research unit Laboratoire d'Etudes en Géophysique et Océanographie Spatiales (LEGOS). This work is also a contribution of the CRATERE project supported by the INSU/LEFE French national program.

**Data Availability Statement:** Publicly available datasets were analyzed in this study. The monthly SLA maps were freely downloaded from the CMEMS portal (http://marine.copernicus.eu; last access on 2 January 2023). The ECMWF's ERA5 reanalysis data were freely downloaded from the Copernicus Climate Change Service portal (https://climate.copernicus.eu; last access on 2 January 2023). Time series of the Niño 3.4 SST anomalies and the AAO index were freely downloaded from NOAA's PSL website (https://psl.noaa.gov/data/climateindices; last access on 2 January 2023). The 15 arc-second gridded bathymetry (GEBCO Compilation Group, 2022) GEBCO_2022 Grid (doi:10.5285/e0f0bb80-ab44-2739-e053-6c86abc0289c) was also freely downloaded from GEBCO's website (https://www.gebco.net; last access on 2 January 2023). The Monthly fields of surface fluxes derived from the NCEP1 and JRA55 products were, respectively, freely obtained from the NOAA's PSL website (https://psl.noaa.gov; last access on 2 January 2023) and the Research Data Archive at the National Center for Atmospheric Research (https://rda.ucar.edu; last access on 2 January 2023). This work also acknowledges the use of gridded Argo data (https://doi.org/10.17882/42182). This data was collected and made available by the International Argo Program and the national programs that contribute to it (https://argo.ucsd.edu; https://www.ocean-ops.org; last access on 2 January 2023). The Argo Program is part of the Global Ocean Observing System.

**Acknowledgments:** The authors thank Matthieu Le Hénaff for his comments and feedback on the original version of the manuscript. The authors also thank two anonymous reviewers for their comments and suggestions that help to improve the manuscript.

**Conflicts of Interest:** The authors declare no conflict of interest.

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
