# Peer review of "Forcing Mechanisms of the Interannual Sea Level Variability in the Midlatitude South Pacific during 2004–2020"

_remotesensing, doi:10.3390/rs15020352_

Round 1

Reviewer 1 Report

The article, Forcing mechanisms of the interannual sea level variability in the midlatitude South Pacific during 2004-2020, study the causes of the observed sea level variability in the midlatitude South Pacific on interannual time scales using altimetry observation and and a 1.5 layer reduced-gravity model. The main result is that the baroclinic Rossby waves forced by the open-ocean wind-stress forcing account for 40 to 75% of the interannual sea level variance in the100°W-180° region at latitudes averaged between34°S-36°S, while the influence of eastern boundary forcing is limited to the region east of 100°W. The manuscript is well structured and comes with understandable and supported conclusions. Apart from some minor suggestions and corrections I think the manuscript is suitable for publication in RS.

Minor comments:

1.     Recently, Sun et al. (2022, https://doi. org/10.1029/2022GL098747) studied the sea level variability in the Southwestern Pacific, with focus on decadal and long-term time scales. It is good to bring this into discussion.

2.     Line 50: I just don’t understand why the steric changes results from water mass redistributions. Steric change is the change of water volume, while the mass redistribution relates to the change of water mass.

3.     Line 53-56: Recent study also found a large contribution of halosteric to the multidecadal sea level change in the southeast Indian Ocean (Lu et al.,2022, DOI: 10.1175/JCLI-D-21-0288.1 ).

4.     Line 307: I suggest to indicate where the Tasman Sea is in the figure.

5.     Figure2: In my view, wind stress curl anomalies over 2017-2018 (Figure 2d) and 2019-2020 (Figure 2f) are very similar in both magnitude and pattern. How to explain their corresponding SLA in Figure 2c and 2d, are totally different.

6.     Figure 1&2: The picture quality needs be improved to better show the maps in Figure 1 and the vectors in Figure2.

7.     Figure9 & 10: For easy reading, it is better to add line legends to the plots.

Reviewer 2 Report

This is a review of the manuscript by Germineaud and collaborators, entitled “Forcing mechanisms of the interannual sea level variability in the midlatitude South Pacific during 2004-2020”, submitted to the journal Remote Sensing, and although the methodology uses data that are essentially from satellite missions (i.e., Jason-3, Sentinel-3A, HY-2A, Saral/AltiKa, Cryosat-2, Jason-2, Jason-1, Topex/Poseidon, ENVISAT, GFO, ERS1/2), the manuscript is 100%  physical oceanography, specifically for the South Pacific Ocean. Using classical methods of physical oceanography, the manuscript focuses on the causes of observed sea level variability in the mid-latitude South Pacific, particularly on interannual timescales, mixing Argos observations, satellites, and atmospheric reanalysis, combined with a 1.5 layer reduced-gravity model.

It is quite interesting that the manuscript demonstrates that the contribution of surface heat fluxes on interannual time scales is considered negligible (as described in its abstract, however … see below). In fact, an excellent introduction, very precise and objective, prepares the reader to understand the discussion of the article. It is still quite curious that the theoretical foundation seems to extinguish any doubts about the regional mechanisms of sea level control, but between lines 99 and 113, the authors reinforce the need for experiments to prove the hypothesis raised, even though throughout the manuscript, several other "doors" are left open for future research.

Although I agree with the method explained between lines 115-120 and beyond, I would like to understand why the authors did not use 1981-2010 as a reference (for example, the DECAV climatology of WOA18) or even the Argo climatology defined between 2005-2017, in order to make it compatible with articles that use these climatology for deseasonalization. Why use the monthly average between 2004-2020 as its own climatology? Note that the use of this time frame contrasts with the subtraction of the GMSL (altimetry) between 1992-2020. Why not use the same reference period for both Argos and Altimetry? I understand that they are different objectives of the manuscript, but why these choices are not clear in the first paragraphs of the methodology, in the way it is written. This must be fixed.

In terms of methodology, although the use of EOS-80 to calculate the parameters used to estimate SSL, TSL and HSL are "classical" in oceanography, I would like to understand why the authors did not use Gibbs' equations, specifically TEOS-10, and in this case, it is necessary to mention about the sensitivity of the EOS-80. There is a vast theoretical framework on the differences between using TEOS-10 instead of EOS-80 for the methodology proposed by the authors. Therefore, bypassing TEOS-10 is not an option in this case. Please justify the use of the EOS-80.

In the introduction section (or perhaps going into section 2. Methods) it would be very interesting for the reader to have a regional map like Figure 2, but showing the bathymetry of the basin, the zones that are described in the introduction, as well as the grid position and the Niño 3.4 area (SST), which are mentioned in the methodology. I believe that the model domain (180° to 70°W, as indicated in line 224) should also be shown in this informative figure. Geographically speaking, this figure facilitates the understanding of the different methods/areas used in the manuscript.

In item 3.1, the explanation that “...the associated southward warming of about 25°S in the Tasman Sea”...”warming was driven by anticyclonic surface wind anomalies in 2015-2016”, needs to be better pointed out in figure 2b and S2b.

Moving forward on the same item, between lines 327 and 335, the authors should better point out WHERE in Figure 2f, the hypothesis that “...there was no weakening or strengthening of the westerly winds in the Southeast Pacific during 2019-2020…” be interpreted. The sequence of explanation versus figures is confusing and makes it difficult to follow.

Throughout item 3, and in particular items 3.1 and 3.2, figures from the manuscript and figures from supplementary material are used. If they (Figures in the supplement) are important for the interpretation of the results, they should not be in supplementary material, do you agree? I also believe that item 3.3 seems "out of place" in relation to the central line of the manuscript, as it specifically focuses on the contribution of Rossby waves. Note that the final sentence “...Despite large uncertainties in the net surface heat flux estimates from atmospheric reanalysis…” ends up being "disapointing" for the reader. I believe this information is important (at least as a warning) and should be included in the methodology or introduction.

I also believe that item 3 is “Results and Discussion”, since item 4 essentially concentrates the conclusions of the manuscript.

Finally, two small suggestions:

Line 39 (suggestion) “the increase in GMSL is caused (to a lesser extent) by the input of….”

Line 119 - Which filter was used? Lanczos ?
